# Characterization of 1,4-Dioxane Biodegradation by a Microbial Community

**Kang Hoon Lee** [1] , **Young Min Wie** [2] **and Yong-Soo Lee** [1,*]

[1]  Department of Civil &Environmental Engineering, Hanyang University, 222 Seongdong-gu, Seoul 04763, Korea; diasyong86@gmail.com

[2]  Department of Materials Engineering, Kyonggi University, Suwon, Gyeonggi-do 16227, Korea; supreme98@kyonggi.ac.kr

*  Correspondence: rokmc907@hanyang.ac.kr; Tel.: +82-2-2220-4633

**Abstract:** In this study, a microbial community of bacteria was investigated for 1,4-dioxane(1,4-D) biodegradation. The enriched culture was investigated for 1,4-dioxane mineralization, co-metabolism of 1,4-dioxane and extra carbon sources, and characterized 1,4-dioxane biodegradation kinetics. The mineralization test indicates that the enriched culture was able to degrade 1,4-dioxane as the sole carbon and energy source. Interestingly, the distribution of 1,4-dioxane into the final biodegrading products were 36.9% into biomass, 58.3% completely mineralized to $CO_2$, and about 4% escaped as VOC. The enriched culture has a high affinity with 1,4-dioxane during biodegradation. The kinetic coefficients of the Monod equation were $q_{max}$ = 0.0063 mg 1,4-D/mg VSS/h, $K_s$ = 9.42 mg/L, $Y_T$ = 0.43 mg VSS/mg 1,4-dioxane and the decay rate was $k_d$ = 0.023 mg/mg/h. Tetrahydrofuran (THF) and ethylene glycol were both consumed together with 1,4-dioxane by the enriched culture; however, ethylene glycol did not show any influence on 1,4-dioxane biodegradation, while THF proved to be a competitive.

**Keywords:** biodegradation; mineralization; kinetic parameter; competitive inhibition

---

## 1. Introduction

1,4-dioxane is a cyclic ether that was commonly used in the past as a stabilizer in chlorinated solvents. It is typically formed as a byproduct during the production of organic fibers from terephthalic acid and ethylene glycol [1–4]. Due to its infinite solubility and high mobility in water with a low Henry's Law ($4.88 \times 10^{-6}$ atm/m$^3$/mol) and low octanol/water partitioning coefficient (log $K_{oc}$ = 1.23), 1,4-dioxane can be easily transported through biological wastewater treatment plants and released to the surface and ground water [2–7]. Since it has carcinogenic properties and recalcitrance in natural water, 1,4-dioxane has been classified as a priority pollutant by the U.S. Environmental Protection Agency (EPA) [8].

The compound is known to biodegrade very slowly [9–13], due to its cyclic structure and ether linkage [14]. In most previous studies, 1,4-dioxane biodegradation was investigated in pure cultures; quite a number of single strains were isolated and characterized by 1,4-dioxane biodegradation [15–23].

However, only few studies investigated 1,4-dioxane biodegradation on mixed cultures [24–26]. Zenker [24] showed the ability to mineralize 1,4-dioxane in the obligate presence of tetrahydrofuran (THF), which is structural analog, as the growth substrate. In the previous study, the activated sludge from polyester manufactures was evaluated for the removal rate of 1,4-dioxane and the results confirmed the dominants species in each sludge sample [27]. The biodegradation potential of 1,4-dioxane was evaluated in different natural bacteria sources, indicating that 1,4-dioxane degrading bacteria are not ubiquitously distributed in natural environment; an interesting conclusion of this study

was that 1,4-dioxane degrading bacteria do not always exist in contaminated sources [28]. These studies have demonstrated the existence of 1,4-dioxane degrading bacteria; however, to date, the kinetics of biodegradation by a mixed culture in the sole carbon source of 1,4-dioxane and the effect of extra carbon source on its kinetic coefficients are sparse [29,30].

The presence of extra carbon sources in raw wastewater may have significant effects on the biodegradation of 1,4-dioxane. Typically, 1,4-dioxnae is known to be formed as a byproduct during the production of organic fibers from ethylene glycol and terephthalic acid [2]; the existence of those materials could inhibit the biodegradation of 1,4-dioxnae.

In this study, a microbial community of bacteria isolated from industrial sludge with 1,4-dioxane was investigated with regard to the biodegradation potential of 1,4-dioxane as the sole carbon source. The distribution of 1,4-dioxane during biodegradation into cell biomass, $CO_2$, and volatile and non-volatile fraction were measured in the mineralization test. The kinetics of 1,4-dioxane biodegradation were estimated alongside with the substrate tolerance of cultures at various concentrations. The co-metabolic degradation of 1,4-dioxane with THF and ethylene glycol was also evaluated in the extra carbon sources effect test.

## 2. Materials and Methods

### 2.1. Enrichment

The industrial sludge was taken from a wastewater treatment plant which has discharged 1,4-dioxane containing wastewater for years. The sludge was transferred to basal salt media (BSM) containing 3240 mg/L $K_2HPO_4$, 1000 mg/L $NaH_2PO_4 \cdot H_2O$, 2000 mg/L $NH_4Cl$, 200 mg/L $MgSO_4 \cdot H_2O$, 12.6 mg/L $FeSO_4 \cdot H_2O$, 3 mg/L $MnSO_4$, 3 mg/L $ZnSO_4 \cdot 7H_2O$, 1 mg/L $CoCl_2 \cdot 6H_2O$ and incubated at room temperature with 200 mg/L of 1,4-dioxane as the sole source of carbon. The culture was washed twice by centrifugation (5000 rpm, 5 min, 20 °C) with sterile BSM containing 100 mg/L 1,4-dioxane as sole carbon source and re-suspended before it was used for inoculation.

### 2.2. Mineralization by Mixed Culture

The mineralization of 1,4-dioxane was investigated to ensure that the mixed culture was able to completely degrade 1,4-dioxane into $CO_2$ and biomass as the final products. A series of 500 mL flasks containing 100 mL of mineral medium were sealed with butyl rubber stoppers. Twenty milliliters of NaOH solution (0.1N) in a 50 mL vial were placed in the flask, as shown Figure 1. Headspace provided oxygen for the mixed culture to degrade 1,4-dioxane by vigorously stirring the medium. Standard HCl solution (0.1N) was used to titrate the amount of NaOH remaining and $CO_2$ was calculated based on the amount of residual NaOH. The test sample was inoculated with the enriched culture and amended with 100 mg/L of 1,4-dioxane. Biotic control was inoculated with the mixed culture without any carbon source and abiotic control amended with 100 mg/L of 1,4-dioxane without inoculation of the mixed culture.

Mass balance of $CO_2$ was computed using the following equation:

$$\text{Recovered } CO_2 \text{ as 1,4-D} = \text{total } CO_2 \text{ absorbed in test sample - } CO_2 \text{ absorbed in biotic control - } CO_2 \text{ produced from cells synthesis} \tag{1}$$

stirring vigorously medium culture:

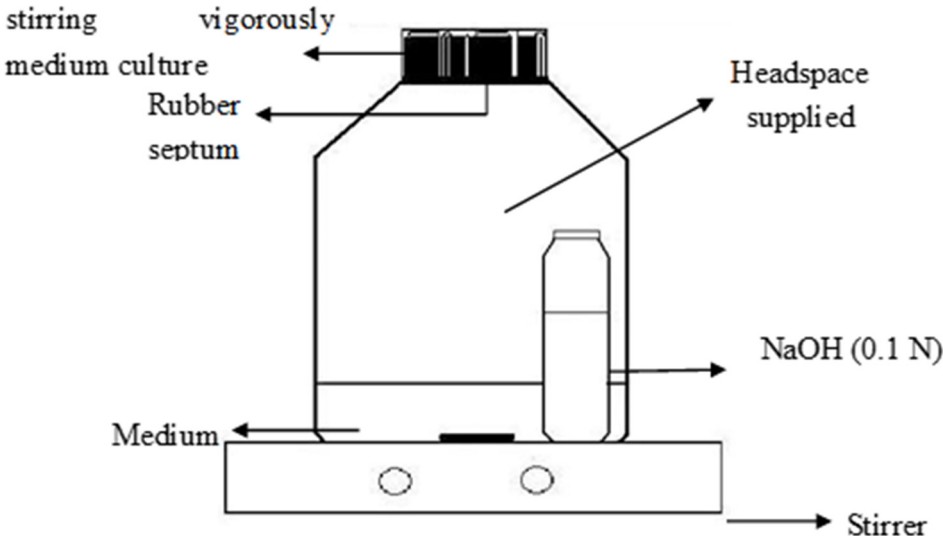

**Figure 1.** Experimental diagram of mineralization test.

*2.3. Biodegradation at Various Initial Concentrations and Degradation Kinetics of 1,4-dioxane*

The experiment was performed in a matrix of 100 mL flasks containing 50 mL of the sample with different concentrations of 1,4-dioxane. The enriched culture was distributed equally in 16 flasks containing 50 mL of BSM, while seven duplicated flasks were amended with different concentrations of 1,4-dioxane as the sole carbon source, ranging from 5 mg/L, 50 mg/L, 200 mg/L, 500 mg/L, 1000 mg/L, 1500 mg/L to 2000 mg/L for the test sample. Another two flasks that were not served 1,4-dioxane were used for biotic control. Two duplicated flasks containing 200 mg/L and 1000 mg/L of 1,4-dioxane were not inoculated with consortium for abiotic controls. All tests were performed in duplicate, incubated at room temperature on a shaker at 160 r/min. Cell production was measured as the difference of MLVSS at the initial and at the end points of the experiment.

*2.4. Effect of Extra Carbon Sources on 1,4-Dioxane Biodegradation*

The test was conducted to investigate the effect of normal carbon source in wastewaters as ethylene glycol and structural analog of 1,4-dioxane as tetrahydrofuran (THF), to biodegradation of 1,4-dioxane by the microbial community. A series of 100 mL flasks containing 50 mL medium were inoculated with 500mg/L of biomass concentration and 100 mg/L 1,4-dioxane. Duplicated flasks were considered as controls with the sole carbon source 1,4-dioxane, while three duplicate flasks containing 1,4-dioxane along with 75 and 150 mg/L of THF were used to evaluate the influence of THF; a further three duplicate flasks were added with 50, 100, 200 mg/L of ethylene glycol to investigate the effect of ethylene glycol. The flasks with 75 and 150 mg/L of THF as a sole carbon source were also investigated for comparison.

*2.5. Analytical Procedures*

The concentration of 1,4-dioxane during incubation was measured by GC-MS (Perkin Elmer Clarus 600) using a liquid–liquid extraction (LLE). The GC/MS analysis was conducted using an Elite-624 column (30 mm × 0.25 mm × 1.4 mm, Perkin Elmer, Waltham, MA, USA) and a helium gas (99.999%, flow of 1 mL/min). The input temperature was set at 240 °C with the initial oven temperature raised to 2400 °C from 400 °C with the total analysis time of 10 min. Soluble COD (chemical oxygen demand) was also measured to double-check the 1,4-dioxane data; a DR/2010 Portable Data Logging Spectrophotometer (HACH) was used. The concentration of bacteria was measured as Mixed Liquor volatile Suspended Solid (MLVSS) according to the standard method.

## 3. Results and Discussions

### 3.1. Mineralization of 1,4-Dioxane

When microorganisms utilize 1,4-dioxane as an electron-donor for synthesis, a portion of its electrons $f_e^0$ is initially transferred to the electron acceptor to provide energy for conversion of the other portion of electrons $f_s^0$ into microbial cells, as illustrated in Figure 2. A part of the electrons in $f_s^0$ are transferred to the acceptor to generate more energy, and another part is converted into a non-active organic cell residue [31] (Bruce E. Rittman and Perry L. McCarty, Environmental biotechnology principles and Applications).

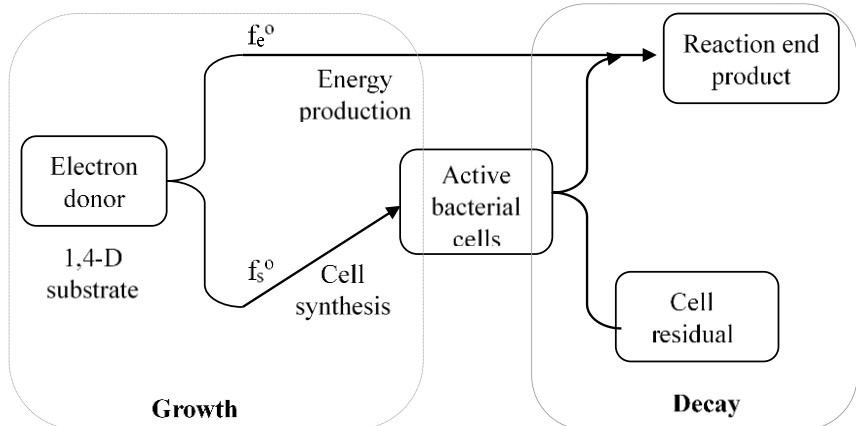

**Figure 2.** Utilization of electron donor for energy production and synthesis.

The mineralization of 1,4-dioxane was investigated to ensure that the mixed culture was able to completely degrade 1,4-dioxane into $CO_2$ and biomass as the final products. The result was interpreted based on $CO_2$ bio-produced and cells associated as an increase in MLVSS.

Carbon dioxide $CO_2$ was produced from the 3 reactions below:

1. Cell synthesis from 1,4-dioxane:

$$3C_4H_8O_2 + 5O_2 + 2NH_3 = 2C_5H_7NO_2 + 2CO_2 + 8H_2O \tag{2}$$

2. Energy production reaction: ($CO_2$ recovered from 1,4-dioxane)

$$C_4H_8O_2 + 5O_2 = 4CO_2 + 4H_2O \tag{3}$$

3. Biomass decay

$$C_5H_7NO_2 + 5O_2 = 5CO_2 + NH_3 + 2H_2O \tag{4}$$

Bio-produced $CO_2$ and $CO_2$ in headspace were driven into the 20 mL of NaOH. For this investigation, the concentration of 1,4-dioxane in liquid portion was almost degraded, $CO_2$ recovered was trapped into NaOH solution. Mineralized $CO_2$ from 1,4-dioxane was equal to the difference between $CO_2$ absorbed in the test samples and biotic control. The fate of 1,4-dioxane during biodegradation was calculated as the average of duplicate tests on the mixed culture grown on 1,4-dioxane as the sole carbon source (Table 1). The final distribution of 1,4-dioxane into the final products derived from biodegradation after 60 h by the microbial community shows that at least 36.9% of 1,4-dioxane was present in the cell mass. The majority of 1,4-dioxane (58.3%) was converted to $CO_2$, while 0.8% remained in the liquid and 4.0% escaped from liquid as volatile organic compounds (VOC). These data document that the enrichment culture can utilize 1,4-dioxane as the sole carbon source and completely oxidize 1,4-dioxane into $CO_2$ and biomass.

**Table 1.** Fate of 1,4-dioxane during biodegradation.

| Content | Values (mg) | Percentage (%) |
|---|---|---|
| Cells associated as 1,4-D | 36.4 ± 2.4 | 36.9 ± 6.6 |
| $CO_2$ recovered as 1,4-dioxane | 57.5 ± 5.4 | 58.3 ± 9.4 |
| Volatile | 3.9 ± 0.1 | 4.0 ± 2.5 |
| 1,4-dioxane residual in liquid | 0.8 ± 0.1 | 0.8 |
| 1,4-dioxane | 98.6 | 100 |

The absence of organic products other than biomass that accumulated in the medium indicates that 1,4-dioxane was completely mineralize by the enriched culture. There was no information on intermediates of 1,4-dioxane degradation, since only 1,4-dioxane pick was found during the analysis. A wide range of 1,4-dioxane initial concentrations was studied in the batch reactors, the degradation and specific degradation rates were found to increase with the increase in the initial concentration within the studied range. The results indicate that 1,4-dioxane was an unlimited substrate for the enriched culture, implying that 1,4-dioxane does not undergo self-inhibition during biodegradation.

### 3.2. Kinetics Parameters Estimation

The batch reactor model was applied to determine the cell yield and endogenous decay rate of the mixed culture during 1,4-dioxane biodegradation. The duplicate abiotic controls were performed to determine the loss of 1,4-dioxane as volatilization; this volatile portion was calculated as 4.35% based on the initial and the end point concentrations of 1,4-dioxane. The cell yield and endogenous decay rates were estimated based on the observation of increase and decrease in the biomass of the tests and biotic control, respectively.

### 3.2.1. Endogenous or Decay Rate

The biotic control was used to determine the decay rate of the mixed culture. Assuming that the loss of active biomass is a first order function, the fraction of inert biomass was ignored.

The rate of endogenous decay was modeled by Equation (5):

$$\mu_{dec} = \left(\frac{1}{X_a}\frac{dX_a}{dt}\right)_{decay} = -k_d \quad k_d = \frac{1}{t}\ln\frac{X_a}{X_0} (*) \tag{5}$$

The experiment was conducted for the initial biomass concentration of 522 mg/L. Figure 3 shows the decrease in biomass concentration after 70 h in the 3-point curve; the negative slope yielded the decay rate coefficient $k_d$. The endogenous-decay coefficient obtained in this study was $k_d = 0.023$ day$^{-1}$.

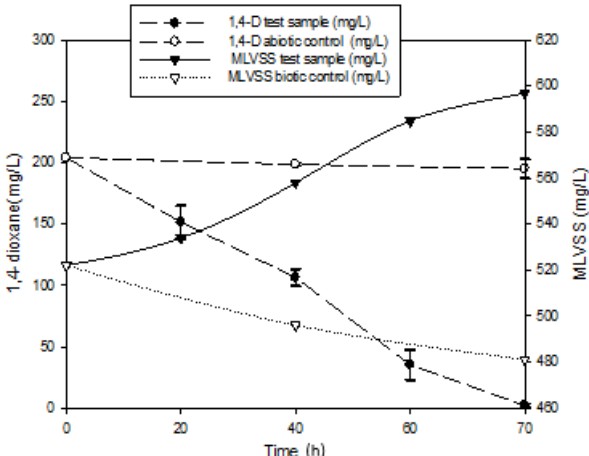

**Figure 3.** Biodegradation of 1,4-dioxane at 200 mg/L of the initial concentration.

### 3.2.2. Observed Cell Yield

The observed cell yield was estimated by linearizing the cell mass increase with 1,4-dioxane consumption (Equation (6)).

$$Y_{obs} = \frac{Cell_{growth}}{1,4-D_{utilization}} \tag{6}$$

In this study, cell yield was estimated based on the observation of biomass production measured relative to the substrate used in the test sample, $Y_{obs}$ = 0.407 mg MLVSS/mg 1,4-dioxane.

The true and the observed yields have the following relationship (Equation (7)):

$$Y_T = Y_{obs}(1 + k_d SRT) \tag{7}$$

By knowing $k_d$ = 0.023 $day^{-1}$, $Y_{obs}$ = 0.407 mg VSS/mg 1,4-dioxane and, in the batch test, SRT = HRT = 70 h; therefore, from Equation (4), $Y_T$ can be calculated as follows: $Y_T$ = 0.434 mg VSS/mg 1,4-dioxane.

### 3.2.3. Biodegradation of 1,4-Dioxane with Different Concentrations and Kinetics

The effect of substrate concentration on microbial community was studied at various initial 1,4-dioxane concentration ranging from 5, 50, 200, 500, 1000, 1500 to 2000 mg/L. All tests were inoculated with the same biomass concentration, VSS = 655 mg/L. The complete degradation of 1,4-dioxane was obtained in all initial concentrations. Figure 4 shows time course variations in 1,4-dioxane at different initial concentrations during the incubation time.

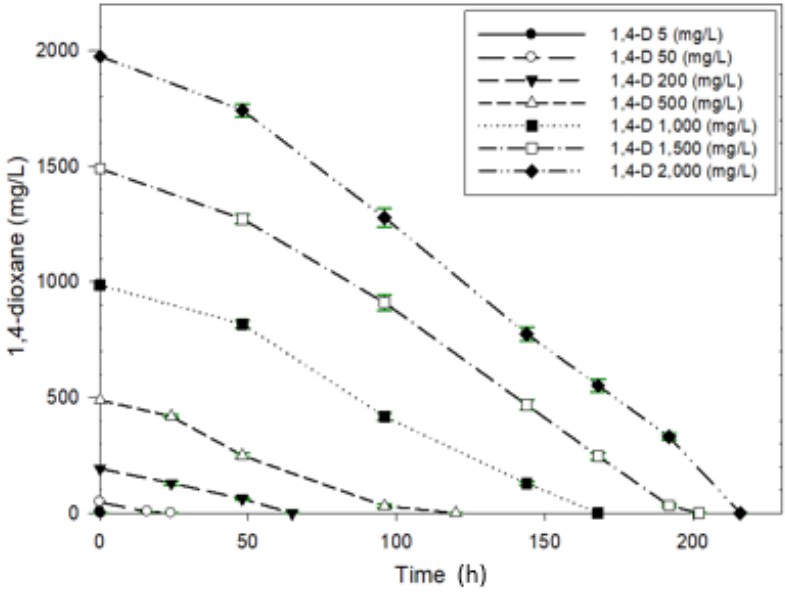

**Figure 4.** Degradation of 1,4-dioxane (1,4-D) at various initial concentrations.

The degradation rates (mg substrate/L/h) at various initial concentrations which were calculated based on the linear slopes are given in Table 2. The degradation rate increased with the increase in initial concentrations, which indicates that 1,4-dioxane was the unlimited substrate for enriched culture. This observation implied that 1,4-dioxane does not undergo self-inhibition during biodegradation.

**Table 2.** Degradation rate at different initial concentrations of 1,4-dioxane.

| 1,4-Dioxane Concentration (mg/L) | Degradation Rate (mg/L/h) |
|:---:|:---:|
| 5 | 1.793 |
| 50 | 2.158 |
| 200 | 3.056 |
| 500 | 4.375 |
| 1000 | 6.167 |
| 1500 | 7.83 |
| 2000 | 9.125 |

The initial specific degradation rates were calculated as the average slopes during first observation of 1,4-dioxane at 48 h with 100, 1500 and 2000 mg/L, at 22 h with 200, 500 mg/L and at 16 h with 50 mg/L. The initial biomass concentration was 655 mg/L, and the increase in biomass was calculated based on the observed cell yield obtained above. The kinetic parameters $K_S$ and $q_{max}$ were determined by fitting a curve to the Monod equation of substrate utilization, as shown in Figure 5.

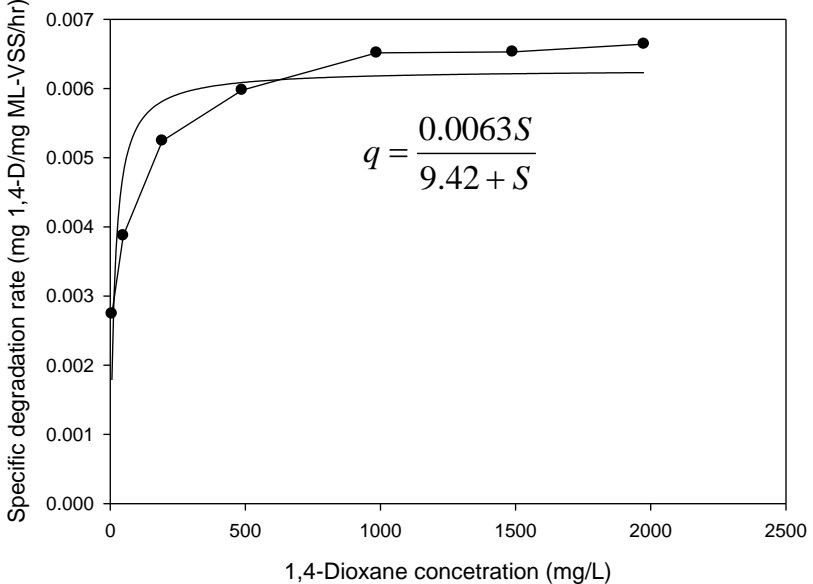

**Figure 5.** Monod plot of 1,4-dioxane degradation at various initial concentrations.

The maximum specific growth rate was as relationship with true yield and substrate utilization rate coefficients:

$$\hat{\mu} = Y_T \hat{q}$$

The kinetic parameters $K_S$ and $q_{max}$ were determined by fitting the curve to the Monod equation of substrate utilization, and Table 3 shows the values of kinetic parameters.

**Table 3.** Kinetic coefficients of 1,4-dioxane biodegradation.

| Kinetic Parameters | Value |
|:---:|:---:|
| $K_s$ (mg 1,4-D/L) | 9.42 |
| $q_{max}$ (mg 1,4-D/mg VSS/h) | 0.0063 |
| $k_d$ (mg VSS/mg VSS/day) | 0.023 |
| $Y_T$ (mg VSS/mg1,4-D) | 0.43 |
| $\mu_{max}$ (day)$^{-1}$ | 0.061 |

The specific degradation rate of 1,4-dioxane in the present study was found to be much lower than those reported in previous studies on pure culture, $q_{max}$ = 1.09 mg 1,4-dioxane/h/mg protein

of *Pseudonocardia dioxanivoran CB 1190* and $q_{max}$= 0.1 mg 1,4-dioxane/h/mg protein of *Pseudonocardia benzenivorans B5* [32], or on mixed culture, $q_{max}$ = 0.45 ± 0.03 mg 1,4-dioxane/mg TSS/day in the presence of THF as growth substrate [33]. This was expected since the mixed culture, including various kinds of bacteria, is able to survive in the presence of 1,4-dioxane; there may be only one or two strains that could use 1,4-dioxane as the carbon source. The $K_S$ obtained in this study was relatively low ($K_S$ = 9.42 mg/L) as compared to those reported in pure culture [32]; $K_S$ = 159 ± 44 mg/L of *Pseudonocardia dioxanivoran* and $K_S$ = 330 ± 82 mg/L of *Pseudonocardia benzenivorans*. These findings could imply that the enriched culture has a high affinity with 1,4-dioxane during biodegradation.

### 3.3. Effects of Ethylene glycol on 1,4-Dioxane Biodegradation

Ethylene glycol was completely degraded at 50, 100 and 200 mg/L after 16, 34 and 45 h, respectively (Figure 6). Moreover, mixed cultures showed the capability of degrading ethylene glycol and 1,4-dioxane at the same time; however, the degrading rate of ethylene glycol substrate was found to be much higher than that of 1,4-dioxane and did not have any influence on 1,4-dioxane substrate utilization (Figure 7).

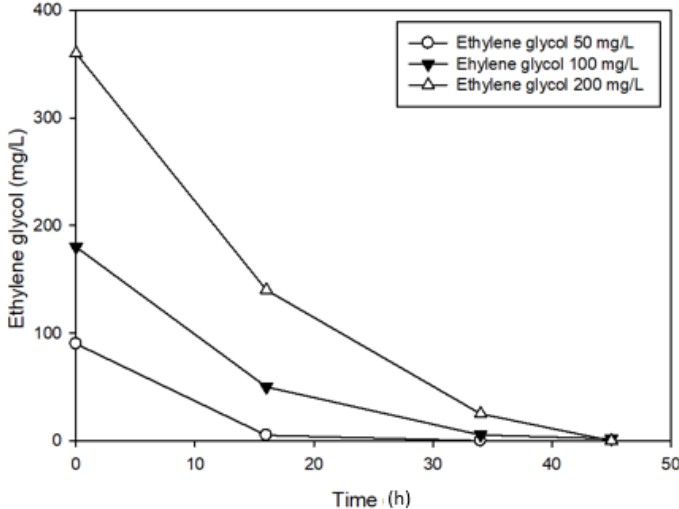

**Figure 6.** Biodegradation of ethylene glycol with various concentration.

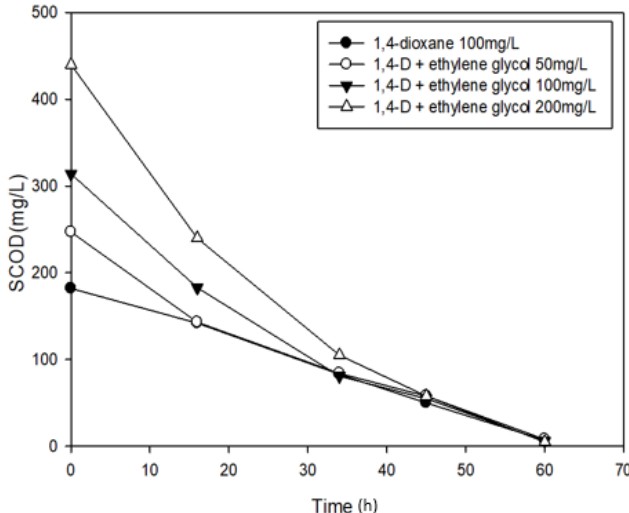

**Figure 7.** Soluble chemical oxygen demand (SCOD) degradation of 1,4-dioxane (1,4-D) and ethylene glycol.

The observation of biomass production indicated that cells grew much faster on ethylene glycol. However, after a complete degradation of ethylene glycol, grown cells did not seem to be helpful for

1,4-dioxane degradation (Figure 8). It may be the case that the bacteria use different enzymes to drive 1,4-dioxane and ethylene glycol into $CO_2$ and biomass. It also could be concluded that, in a microbial community, including various kinds of bacteria, there may only be some strains that are able to use 1,4-dioxane directly, while others mineralize the by-products of 1,4-dioxane biodegradation into $CO_2$. In this case, the strains able to degrade 1,4-dioxane also use ethylene glycol; however, grown cells lose their capability of degrading 1,4-dioxane.

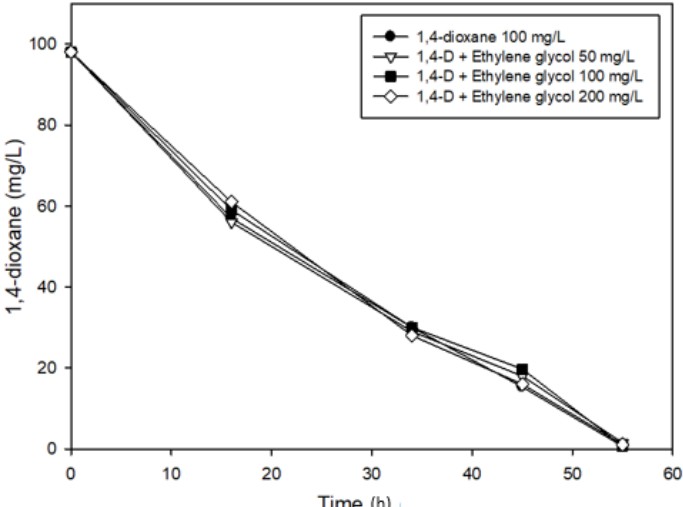

**Figure 8.** Biodegradation of 1,4-dioxane (1,4-D) in the presence of different ethylene glycol concentrations.

### 3.4. Effects of THF on 1,4-Dioxane Biodegradation

1,4-dioxane and THF were degraded together at all tested concentrations. However, Figure 9 indicates that THF initially inhibited the degradation of 1,4-dioxane. The initial degradation rate of 1,4-dioxane was negatively influenced at a higher concentration of amended THF. Moreover, biodegradation of THF as the sole carbon source showed a higher degradation rate (Figure 10). The enriched culture was able to completely degrade 150 mg/L THF as the sole carbon source after 45 h, while 60 h were needed for the complete degradation of 100 mg/L 1,4-dioxane as the sole carbon source of the same initial biomass concentration. In addition, THF biodegradation was not affected by the presence of 1,4-dioxane. The obtained results were used to estimate the kinetic parameters of 1,4-dioxane as the sole carbon source and THF as the sole carbon source together with 1,4-dioxane and THF cometabolism as COD measurements using the Monod equation of the batch test model.

The results presented in Table 4 show that the maximum specific degradation rate of THF was double that of 1,4-dioxane (0.0088 mg COD(1,4-D)/mg-MLVSS/h compared to 0.0167 mg COD (THF)/mg-MLVSS/h), and that the $K_S$ of THF biodegradation is also lower than the $K_S$ of 1,4-dioxane biodegradation. This implied that enriched culture has a higher affinity to degrade THF than 1,4-dioxane. With an increase in the THF concentration from 75 mg/L to 150 mg/L, the maximum specific degradation rate decreased from 0.0171 to 0.0149 mg COD/mg-MLVSS/h. This provides a strong evidence for the conclusion that 1,4-dioxane and THF inhibit each other in cometabolism by the enriched culture.

In previous reports, THF was considered as the bio-stimulation of 1,4-dioxane biodegradation or the growth substrate for bacteria that are able to degrade 1,4-dioxane. Zenker [24] indicated that the biodegradation of 1,4-dioxane by an isolated consortium from soil microcosms is dependent on THF, and that 1,4-dioxane degradation could not be found in the absence of THF. However, in this study, THF has been proven to be a competitive inhibition of 1,4-dioxane. This is expected, since THF has a similar structure to 1,4-dioxane, using the same degradative enzymes to drive into $CO_2$ and biomass.

The mechanism of competitive inhibition is that the structural analog binds the catalytic side of the degrading enzyme.

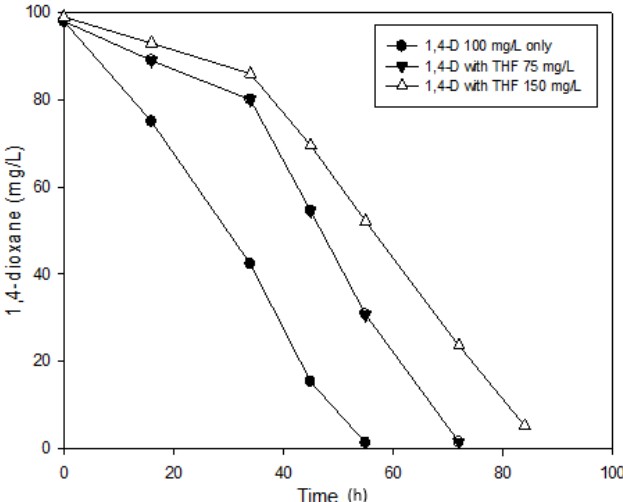

**Figure 9.** The biodegradation of 1,4-dioxane (1,4-D) in effect of different concentrations of tetrahydrofuran (THF).

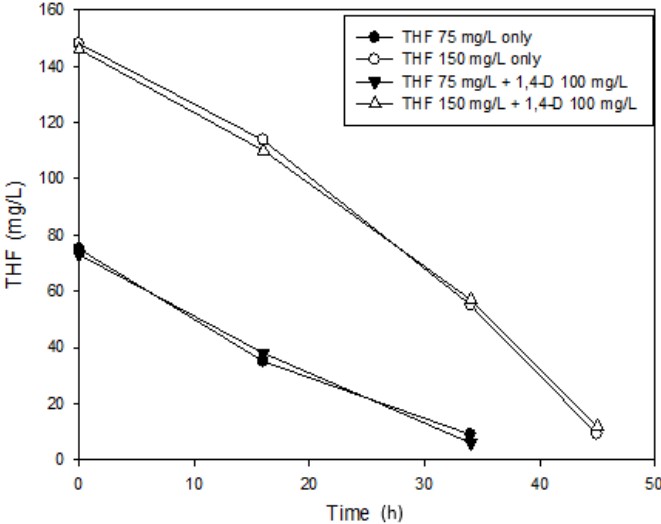

**Figure 10.** The biodegradation of THF in effect of 1,4-dioxane (1,4-D).

**Table 4.** Kinetic coefficients of cometabolism 1,4-dioxane and THF as COD at different concentration of THF.

| Kinetic Coefficients | 1,4-Dioxane as Sole Source | THF as Sole Source | 1,4-D&THF 75 mg/L | 1,4-D&THF 150 mg/L |
|---|---|---|---|---|
| $q_{max}$ (mg COD/mgVSS/h) | 0.0088 | 0.0167 | 0.0171 | 0.0149 |
| $K_{eff}$ (mg/L) | 9.08 | 6.4 | 131.72 | 345.6 |

## 4. Conclusions

The enriched culture was able to degrade 1,4-dioxane as the sole carbon and energy source. The distribution of 1,4-dioxane during biodegradation was 36.9% into biomass, 58.4% completely mineralized to $CO_2$, and about 4% was loss as volatilization.

The kinetic coefficients of the Monod equation were estimated, and the results indicate that the microbial community of bacteria has a high affinity with 1,4-dioxane.

The additions of ethylene glycol and THF at different concentrations were to evaluate the influence of extra carbon sources on 1,4-dioxane biodegradation. Ethylene glycol was consumed together with 1,4-dioxane by enriched culture, but did not have any influence on 1,4-dioxane biodegradation. While THF was proven to be a competitive inhibitor of 1,4-dioxane.

**Author Contributions:** Conceptualization, K.H.L.; Methodology, K.H.L.; Software, K.H.L. and Y.M.W.; Validation, Y.-S.L.; Investigation, K.H.L.; Data Curation, Y.-S.L.; Writing—Original Draft Preparation, K.H.L.; Writing—Review and Editing, Y.-S.L.; Visualization, Y.M.W.; Supervision, Y.-S.L.; Project Administration, Y.-S.L. All authors have read and agreed to the published version of the manuscript.

**Funding:** This research received no external funding.

**Acknowledgments:** This work was supported by Korea Environment Industry and Technology Institute (KEITI) through Environmental R&D Project on the Disaster Prevention of Environmental Facilities Project, funded by Korea Ministry of Environment (MOE) (2020002870004).

**Conflicts of Interest:** The authors declare no conflict of interest.

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
