# Peer review of "Characterization of 1,4-Dioxane Biodegradation by a Microbial Community"

_water, doi:10.3390/w12123372_

Round 1

Reviewer 1 Report

The paper describes the influence of a Characterization of 1,4-dioxane Biodegradation by  Mixed Consortium of Bacteria. I think this paper needs major revision in order to be accepted as a journal paper. Followings are detailed comments.

  1. The study lacks a clearly defined research goal.
  2. Only the research results are presented in the paper. The Discussion chapter is missing.
  3. „Mixed consortium of bacteria” should be changed by „microbial community”
  4. Lack of microbial community characteristics, method of isolation, species participation, etc.
  5. Minor errors should be corrected:

Fig. 1 01N should be changed by 0.1 N

Line 39.  atm. It is not an SI unit. etc.

Author Response

The paper describes the influence of a Characterization of 1,4-dioxane Biodegradation by Mixed Consortium of Bacteria. I think this paper needs major revision in order to be accepted as a journal paper. Followings are detailed comments.

1.The study lacks a clearly defined research goal.

Response 1: Thanks for your advice. I’ve added clearly research goal on line 50

2.Only the research results are presented in the paper. The Discussion chapter is missing.

Response 2: Discussions were in the result chapter. I changed the name of chapter “Results and Discussions”

3.„Mixed consortium of bacteria” should be changed by „microbial community”

Response 3: I’ve changed all of “Mixes consortium of bacteria” to “Microbial community”

4.Lack of microbial community characteristics, method of isolation, species participation, etc.

Response 4: I’ve tried to isolate microbial community almost 2 years but we failed it. There must be some kinds of mechanism of bacteria degrading 1,4-dioxane that we can’t find. We are still working on it. Once we finished the isolation, new article will be prepared.

5.Minor errors should be corrected:

Fig. 1 01N should be changed by 0.1 N

Line 39. atm. It is not an SI unit. etc.

Response 5: Thanks for your correction. I’ve changed to “0.1 N”

Reviewer 2 Report

The submitted manuscript is interesting and contain many valuable results, however, some parts of the study are unclear and have to be explained.

Abstract:

1) The term "enriched culture" is quite unclear. Could the Authors how they understand it?

Materials and methods:

2) How were the microorganisms added to the dioxane culture? Have they been centrifuged and washed? If activated sludge or sewage was added directly, how much? In this case, it should also be remembered that the added solution also contains organic matter, so dioxane will not be the only source of carbon. 

3) What kind of bacteria were present? Were there only bacteria in consortium? In raw WWTP activated sludge are present also other microorganisms. 

4) Line 78: I have doubts about this equation. I would like you to explain the two terms of the equation that are subtracted. How is CO2 from cell synthesis determined and why is it subtracted? If 1,4-dioxane was the only source of carbon, then the synthesis of compounds that make up cells to some extent will also come from it.

5) Moreover, the subtraction of values ​​in the biotic control is not entirely correct, in my opinion, the addition of dioxane most likely slowed down the metabolism of some microorganisms in the consortium, and thus it can be assumed that the respiratory activity of cells in the dioxane test will be analogous to that in the biotic test without this chemical. Figure 2 does not help so much, because it does not include carbon balance and the emission of CO2. However, there are three equation below Fig. 2, but they do not correspond with equation in line 78, which contain the part about cells synthesis but not their decay.

6) Line 91: The choice of additional carbon source should be explained. Why exactly such compounds? Why could they support biodegradation of 1,4-dioxane? The similarity of THF molecule is not so sufficient. Is there any probability of the presence of these compounds along with 1,4-dioxane?

7) How do the Authors understand the COD? How was it measured?

Results:

8) Table 1: Statistical errors or standard deviations should be included. Moreover, I would remove the first row or I would move it to the bottom of table to show that the values above give together 100%.

9) Table 3: What culture (what 1,4-D concentration) is it reffered to?

10) the degradation curves of ethylene glycol and THF should be also shown. If I see properly, the manuscript contain only the graphs describing the 1,4-dioxane degradation.

Author Response

The submitted manuscript is interesting and contain many valuable results, however, some parts of the study are unclear and have to be explained.

Abstract:

1) The term "enriched culture" is quite unclear. Could the Authors how they understand it?

Response 1: Since industrial sludge contains various types of bacteria so we need to enrich the culture by giving 1,4-dioxane as sole carbon source.

Materials and methods:

2) How were the microorganisms added to the dioxane culture? Have they been centrifuged and washed? If activated sludge or sewage was added directly, how much? In this case, it should also be remembered that the added solution also contains organic matter, so dioxane will not be the only source of carbon.

Response 2: Thank you for your advice. I’ve added more details about culture preparation in line 67-69.

3) What kind of bacteria were present? Were there only bacteria in consortium? In raw WWTP activated sludge are present also other microorganisms.

Response 3: As you know there are numerous kinds of bacteria in WWTP sludge. In enrichment process, 1,4-dioxane was the only carbon source for sludge more than 6 months, so that we could assume that every other organisms except 1,4-dioxane degrading bacteria are negligible.

4) Line 78: I have doubts about this equation. I would like you to explain the two terms of the equation that are subtracted. How is CO2 from cell synthesis determined and why is it subtracted? If 1,4-dioxane was the only source of carbon, then the synthesis of compounds that make up cells to some extent will also come from it.

Response 4: Of course there could be synthesis of compounds that make up cells in the flask. However, in this session, experiment was conducted to estimate the CO2 produced in the process of 1,4-dioxane mineralization not from cell synthesis.

5) Moreover, the subtraction of values ​​in the biotic control is not entirely correct, in my opinion, the addition of dioxane most likely slowed down the metabolism of some microorganisms in the consortium, and thus it can be assumed that the respiratory activity of cells in the dioxane test will be analogous to that in the biotic test without this chemical. Figure 2 does not help so much, because it does not include carbon balance and the emission of CO2. However, there are three equation below Fig. 2, but they do not correspond with equation in line 78, which contain the part about cells synthesis but not their decay.

Response 5: Like in Response 3, we assumed that only 1,4-dioxane degrading bacteria are activated, so we should subtract CO2 absorbed in biotic control.

6) Line 91: The choice of additional carbon source should be explained. Why exactly such compounds? Why could they support biodegradation of 1,4-dioxane? The similarity of THF molecule is not so sufficient. Is there any probability of the presence of these compounds along with 1,4-dioxane?

Response 6: Previous researches (Zenker, Mahendra, Bernhardt, Sei and more) had claimed the importance of presence of THF. It could affect to 1,4-dioxane biodegradation either in positively or not.

The reason of ethylene glycol is in line 51-52.

7) How do the Authors understand the COD? How was it measured?

Response 7: Analytical Method section has been changed. (line 113-115)

Results:

8) Table 1: Statistical errors or standard deviations should be included. Moreover, I would remove the first row or I would move it to the bottom of table to show that the values above give together 100%.

Response 8: Table 1 has been revised. Thank you.

9) Table 3: What culture (what 1,4-D concentration) is it reffered to?

Response 9: It is mixed consortium of bacteria with 200 mg/L of 1,4-dioxane.

10) the degradation curves of ethylene glycol and THF should be also shown. If I see properly, the manuscript contain only the graphs describing the 1,4-dioxane degradation.

Response 10: The degradation curves of ethylene glycol has been added. Fig 10 shows the degradation curves of THF. Thank you so much.

Round 2

Reviewer 1 Report

The paper can be accepted for publication.

Reviewer 2 Report

Thank you very much for your kind responses. I think that now the manuscript is suitable for publication.